# Executive Profile of the Logopenic Variant of Primary Progressive Aphasia: Comparison with the Semantic and Non-Fluent Variants and Alzheimer’s Disease

**DOI:** 10.3390/brainsci13030406

**Published:** 2023-02-26

**Authors:** Sandrine Basaglia-Pappas, Bernard Laurent, Jean-Claude Getenet, Anne Boulangé, Aurelia Rendón de laCruz, Isabelle Simoes Loureiro, Laurent Lefebvre

**Affiliations:** 1Department of Neurology and Neuropsychology, Memory Center, North Hospital, University Hospital Center, CEDEX 2, 42055 Saint-Etienne, France; 2Department of Cognitive Psychology and Neuropsychology, University of Mons, 7000 Mons, Belgium

**Keywords:** primary progressive aphasia, Alzheimer’s disease, executive functions, assessment, diagnosis

## Abstract

The logopenic variant of primary progressive aphasia (lvPPA) shows different features from the non-fluent (nfvPPA) and semantic (svPPA) variants of PPA. Although language impairments remain the core symptoms, studies have highlighted the presence of executive disorders at the onset of the disease. Nevertheless, the results are contradictory, particularly in lvPPA. The aim of this study was to explore the executive profile of lvPPA. We compared executive functioning in lvPPA with the other two variants of PPA, Alzheimer’s disease (AD) and a cognitively healthy group. In total, 70 patients with PPA, 32 patients with AD, and 41 healthy controls were included. They underwent a comprehensive executive battery assessing short-term and working memory, inhibition, flexibility, planning, and initiation. The analyses showed significant differences between the lvPPA group and the control group, except on visuospatial spans and the Stroop test, and between the lvPPA group and the other PPA groups and the AD group for several tasks. Thus, this research highlighted the existence of an executive dysfunction from the onset of the disease in lvPPA but also in the other two variants of PPA.

## 1. Introduction

Primary progressive aphasia (PPA) identifies a clinical neurodegenerative syndrome, with the earliest symptom being a decline in speech and language. Knowledge of the clinical profile of PPA continues to evolve and become more accurate.

A progressive disorder of language associated with the atrophy of the frontal and temporal regions of the left hemisphere was first described by Pick in 1892 [1]. Almost a century later, Mesulam reported patients with “slowly progressive aphasia” in 1982 [2]. PPA identifies a group of neurodegenerative syndromes that selectively impair language processing, with the relative preservation of cognitive skills for at least two years after the onset of the disease [2,3,4,5]. During this time, the impact on the instrumental activities of daily living is entirely due to language difficulties [4]. The international consensus criteria have given rise to published recommendations for the diagnosis and classification of PPA [3]. They defined three clinical variants based on core impairments and associated features [3]: non-fluent/agrammatic (nfvPPA), semantic (svPPA), and logopenic (lvPPA) variants. First, nfvPPA is characterized by the agrammatism and/or apraxia of speech. A diagnosis of svPPA is established when anomia, impaired word comprehension, and impaired object recognition are present. Moreover, lvPPA is characterized by word retrieval difficulties and impaired repetition due to a verbal short-term memory deficit. Classifying the different variants of PPA is complex, and several authors have presented algorithms to diagnose cases of PPA based on key clinical features [6,7,8]. However, classifications can still be difficult. Neuroimaging features support the clinical diagnosis, revealing bilateral but asymmetric atrophy: left greater than right posterior fronto-insular atrophy in the non-fluent variant, left temporoparietal atrophy in the logopenic type, and anterior temporal atrophy in the semantic variant [3,9]. In this clinical syndrome, neuropathology is heterogeneous, with 60% of patients with PPA presenting frontotemporal lobar degeneration (FTLD), and the remaining 40% with Alzheimer’s disease (AD) pathology [5]. The most frequent associations have been reported between FTLD-nfvPPA patients and tauopathy, FTLD-svPPA patients and ubiquitin/TDP43 proteinopathy, and lvPPA patients and AD pathology [3,5,10,11].

Referring to the clinical syndrome, AD is the most common neurodegenerative disease, accounting for an estimated 60% of all neurodegenerative pathologies [12]. Regarding cognitive disorders, at the early stage of the disease, an impairment in episodic memory is considered to be the core symptom [13]. For several decades, studies have highlighted executive function (EF) disorders in AD [14]. Indeed, a cognitive dysexecutive syndrome is described in more than 75% of AD cases [15]. 

EFs are defined as high-level cognitive abilities [16]. In this study, we focus on the most widely accepted ones: cognitive inhibition (the ability to suppress distracting information or a dominant answer), cognitive flexibility (the ability to quickly and flexibly adapt to changed circumstances), planning (the ability to organize a series of thoughts or actions into an optimal sequence to achieve a goal), initiation (the capacity to initiate a mental activity), and the executive part of the working memory (a short-term memory system for processing verbal and nonverbal information) [17,18,19]. In terms of neuroanatomy, EFs are linked to the prefrontal cortex (PFC), with its four areas: dorsolateral, orbitofrontal, ventromedial, and anterior cingulate cortices. The PFC is connected to all associative cortex areas, which receive sensory input, allowing for the multimodal integration of the surrounding environment [17,20]. 

At the onset of PPA, EFs are believed to remain relatively unaffected [4,21]. However, researchers have recently been investigating functions other than language [7,22,23,24,25,26,27], and specifically EFs [23,28,29,30,31,32,33,34,35,36,37]. The reported data show an impairment of several EFs, including cognitive flexibility and inhibition [38] in nfvPPA, and cognitive flexibility [39] in svPPA. In lvPPA, the main subject of this study, executive dysfunction has been described [7,40,41]. Impaired verbal short-term memory, often investigated with a forward span task, constitutes a major feature, which is widely described in the literature [30,31,32,35,40,42,43,44,45]. A verbal working memory impairment, often evaluated with a backward span task, is also reported in lvPPA [22,34,42,46,47,48]. In the early stages, a dissociation has been noted between verbal short-term memory performance, which is impaired, and visuospatial short-term memory, which is less deficient and sometimes preserved, especially in comparison with patients with AD [34,35,43,49,50], as this function is subserved by the right parietal and frontal regions [51]. A recent study also described the preservation of visuospatial short-term and working memory [36]. However, some disagreement exists. Certain authors have reported a significant impairment in visuospatial short-term and working memory (often assessed with forward and backward spans, respectively) [31,32,40]. Inhibition, assessed with the Stroop test in a recent study of 36 lvPPA patients, is also impaired [36]. Information processing speed (assessed with the Trail Making Test A) and flexibility (assessed with the Trail Making Test B) are altered as well [29,32,34,35,36,42,47,49]. Initiation, assessed by design fluency, has been described as preserved [27,42,52]. Finally, a recent study [36] highlighted a planning impairment, assessed by the Tower of London test.

However, these studies are difficult to compare as the authors did not always use the same tasks to investigate the assessed functions. Thus, the results of these studies are often contradictory [41]. Moreover, only a few studies have investigated several EFs together [27,29,36,53]. Therefore, it is difficult to definitively conclude that an executive dysfunction exists in PPA, and especially in lvPPA [53].

If we now compare lvPPA to AD, as Rahul and Ponniah [54] remind us, lvPPA might represent the “uni-hemispheric” form of AD [50]. Recent neuropathological evidence has indicated that lvPPA may be associated with AD [5,6,35,42,48,50,55,56,57]. As for cognitive functions, lvPPA is associated with AD in that it also involves episodic memory impairment [10,50]. However, lvPPA and AD differ in their effects on working memory. Indeed, although verbal memory is impaired in both pathologies [31,47,48], a greater deficit is found in visuospatial span tasks than in verbal span tasks in patients with AD, compared with lvPPA patients; this deficit is linked to the atrophy of the parieto-occipital associative regions [31]. Visuospatial short-term memory and working memory are described as less impaired in lvPPA, which may help with differential diagnosis [36]. However, these claims are controversial [31,32].

The objectives of our study were twofold. First, we sought to provide new insights into EF impairment or preservation in lvPPA, i.e., the variant that has been least often studied in the literature [41]. Indeed, new measures are needed to improve diagnosis accuracy. Second, we sought to compare the executive profile of lvPPA with those of the two other variants of PPA and with the AD profile, in order to contribute to the early differential diagnosis of these three conditions and to the planning of appropriate interventions [25,58]. 

## 2. Materials and Methods

### 2.1. Population

To carry out this research, five groups with a total of 143 participants were included. First, over a three-year period, a sample of individuals with PPA was recruited, especially at the Research, Resources, and Memory Center at the North University Hospital Center, Saint-Etienne, France. A few participants were also recruited at Purpan Hospital, Toulouse, and Beauvais Hospital. In total, 22 patients with lvPPA, 22 patients with nfvPPA, and 26 patients with svPPA were recruited.

Inclusion criteria for the study were a clinical diagnosis of PPA, a consultation by a neurologist at an investigation center for cognitive difficulties, a magnetic resonance image within six months of initial evaluation, and agreement to participate in the study after reading the information letter.

All PPA participants met the criteria for PPA [3]. They underwent a neurological examination and a cognitive speech and language assessment, as well as cerebral medical imaging underpinning the clinical diagnosis and biomarker investigation. In order to have the purest possible clinical PPA syndromes, i.e., without associated disorders, several patients were not included in the study: four with an atypical parkinsonian syndrome (with mainly tremors of the right upper limb) and two with liquid swallowing disorders.

The mean age was 68.95 (±9.3) for patients with lvPPA, 69.55 (±4.8) for those with nfvPPA, and 66.27 (±10.2) for those with svPPA. The mean durations of education in years were 11.59 (±2.5), 11.68 (±2.8), and 12.42 (±2.7) for the three groups, respectively. Statistical analyses showed no significant difference between groups for age and education level. The duration of the disease at the time of testing varied from 23 to 31 months for all PPA groups. 

We also recruited 32 individuals with early-stage AD (MMSE scores above 20 out of 30) at the Research, Resources, and Memory Center in North University Hospital Center. Their mean age was 65.66 (±9.5) and the mean education duration was 11.69 (±2.7) years.

Forty-one education- and age-matched healthy volunteers were recruited. Their mean age was 65.73 (±10.1) and the mean education duration was 12.07 (±3.7) years. To be included, they had to present no history of neurological or psychiatric disorder and no cognitive complaints.

Three patients and two control participants were left-handed.

Demographic data on the participants and their scores on the Mini-Mental State Examination (MMSE) [59], a widely used cognitive screening tool, are presented in Table 1.

The study was conducted according to the guidelines of the Declaration of Helsinki and approved by the University of Mons ethics committee. All participants (patients and controls) gave written informed consent for their participation. All of them agreed to take part in the study. Data were anonymized before the statistical analyses were completed.

### 2.2. Methods

All participants underwent neuropsychological testing with a comprehensive battery of executive function tasks, which included the verbal forward and backward digit span subtest from the WAIS-IV [60] and the visuospatial forward and backward digit span subtest from the WMS-III [61] to assess short-term and working memory; the Trail Making Test (TMT) [62] part A to test the processing speed and part B to examine the efficiency of cognitive flexibility; the color word Stroop test [63] to assess inhibition levels; the Tower of London test [64] to assess planning; and the design fluency Ruff Figural Fluency Test (RFFT) [65] to assess initiation levels. Thus, the protocol included classic tasks, such as the TMT and the color word Stroop test, but also less commonly administered ones, such as the Tower of London test and the design fluency RFFT. Because most tasks depend on verbal instructions, verbal responses, or covert verbal reasoning, we first verified that patients understood the instructions (especially for the Tower of London test), knew the alphabet well, could name colors, and could repeat numbers.

Statistical analyses were conducted using IBM SPSS version 25.0. Prior to conducting these analyses, all variables were checked for normality of distribution using Shapiro–Wilk test. Since variables were non-normally distributed, group comparisons of demographic data were carried out with nonparametric tests and chi-squared test for gender. Group comparisons on neuropsychological measures were investigated using Kruskal–Wallis tests and Mann–Whitney tests. A *p* value of < 0.05 was adopted to determine the statistical significance.

## 3. Results

### 3.1. Demographic Data Comparison

Group comparisons of the demographic data revealed no significant difference between groups regarding age, gender, laterality, and years of education.

### 3.2. Group Comparison to Controls

Scores for each group and between-group differences in EF measures are presented in Table 2, with descriptive results as means and standard deviations.

Relative to healthy controls, the lvPPA group showed significantly poorer scores than the control group for most tasks (*p* < 0.05), except the forward visuospatial span (*U* = 199.500; *z* = −1.859; *p* = 0.063), backward visuospatial span (*U* = 257.500; *z* = −0.605; *p* = 0.545), and Stroop interference tasks (*U* = 156.500; *z* = −1.903; *p* = 0.057). Patients in the nfvPPA and AD groups had significantly lower scores for all the tasks than controls (*p* < 0.0001). The svPPA group scored significantly lower than controls only on the RFFT design fluency task (*U* = 284.500; *z* = −2172; *p* = 0.03).

### 3.3. Clinical Groups Comparisons

In comparing the lvPPA group to other clinical groups, patients with nfvPPA showed significantly poorer scores for the following tasks: backward digit span (*U* = 86.500; *z* = −2.301; *p* = 0.021), forward visuospatial span (*U* = 14.000; *z* = −4.333; *p* < 0.0001), backward visuospatial span (*U* = 62.500; *z* = −2.334; *p* = 0.020), Stroop interference (*U* = 41.000; *z* = −2.555; *p* = 0.011), and RFFT design fluency (*U* = 39.000; *z* = −2.162; *p* = 0.031). The forward digit span, TMT A and B, and Tower of London tasks did not distinguish between the two groups (*p* > 0.05). The svPPA group performed significantly better than the lvPPA group on the TMT A (*U* = 35.000; *z* = −3.950; *p* < 0.0001), TMT B (*U* = 72.000; *z* = −2.788; *p* = 0.005), forward digit span (*U* = 98.500; *z* = −2.695; *p* = 0.007), and backward digit span (*U* = 99.500; *z* = −2.566; *p* = 0.010) tests. Analyses did not reveal any significant difference for the other tasks. Finally, patients with AD performed significantly worse on only the forward visuospatial span (*U* = 22.500; *z* = −4.652; *p* < 0.0001) and backward visuospatial span (*U* = 17.500; *z* = −3.817; *p* < 0.0001) tasks. The other tasks did not differentiate the two groups (*p* > 0.05).

As for other between-group differences, participants in the nfvPPA group performed significantly worse than participants in the AD group on the forward digit span (*U* = 94.500; *z* = −3.178; *p* = 0.001) and backward digit span (*U* = 125.000; *z* = −2.330; *p* = 0.020) tests. The analyses did not show any significant difference between the two groups for the other tasks (*p* < 0.05). The AD group scored significantly lower than the svPPA group on all tasks (*p* < 0.05). Finally, the analyses showed a significant difference between the nfvPPA and svPPA groups for all tasks (*p* < 0.05). Participants in the nfvPPA group performed significantly worse than participants in the svPPA group. 

To summarize, between-group comparisons showed that all three PPA groups presented significant executive impairments compared with healthy controls. Among PPA groups, the nfvPPA and svPPA patients differed most on EF tests, followed by the svPPA and lvPPA groups. The nfvPPA and lvPPA groups did not differ significantly on the EF tests.

Figure 1 presents the executive tasks, showing statistically significant differences between the lvPPA group and other clinical groups.

## 4. Discussion

The main objective of this study was to explore the executive profile of lvPPA at the early stage of the disease and thus better characterize patients’ difficulties. Indeed, few studies have studied several EFs at the same time. We also wished to clarify the conflicting results reported in previous studies. Furthermore, we wanted to compare executive functioning in lvPPA with such functioning in the other two variants of PPA and in AD patients to help with differential diagnosis.

### 4.1. lvPPA and Control Groups

In the comparison of the lvPPA and control groups, the analyses showed a deficit in verbal working memory, as the lvPPA group scored significantly lower than the control group (*p* = 0.007). Patients did not score much lower than the control group in inhibition levels, as described in the literature [36,41]. A longitudinal study [41] reported impaired performance on the Stroop test in a patient who had been diagnosed three years earlier. The more sensitive Stroop test revealed an impairment in this patient, whereas the easier TMT showed preserved performance, emphasizing the usefulness of longitudinal follow-up before one can conclude that EFs are preserved. Thus, the good results in our study reflect the variation in executive function performances at individual and group levels reported in these patients [47,48]. An inhibition impairment, evaluated with the Stroop test, seems to appear rapidly as the disease develops. On planning, assessed by the Tower of London test, lvPPA patients scored significantly lower than the control group. These results are consistent with a recent study that describes a planning impairment in lvPPA [36]. As for initiation, evaluated by RFFT design fluency, the results indicate that this EF is deficient in lvPPA participants compared to controls (*p* = 0.010), and lvPPA patients performed poorly on initiation, which has not been reported in other studies [27,52]. To conclude, the lvPPA group showed moderately impaired EFs, as described in the literature [27], with preserved inhibition and visuospatial short-term and working memory.

### 4.2. nfvPPA and Control Groups

The nfvPPA group’s results are consistent with recent studies identifying executive dysfunction in nfvPPA [23,33,36,40,41], contrary to the initial descriptions by Mesulam [4,66], who claimed that there was no dysexecutive syndrome at the onset of the disease. Verbal and visuospatial short-term and working memory, inhibition, processing speed, cognitive flexibility, initiation, and planning are not efficient, even at the onset of nfvPPA. With regards to initiation, some authors [67] describe a deficit, while others [28] say that this function is preserved at the early stage, if the damage remains limited to the inferior frontal gyrus, insofar as this type of task involves more right frontal circuits. Poor results in the RFFT design fluency task showed that the atrophy progressively spreads to the dorsolateral prefrontal regions. Stuss [20] stated that an initiation impairment is linked to the frontal lobes [17]. The nfvPPA participants therefore presented difficulties in initiating and maintaining control strategies in the design fluency task over time, leading to a lower number of productions than in controls. To sum up, the nfvPPA group showed major EF difficulties. Thus, their language deficits were not isolated. Numerous studies have recently shown this twofold alteration [23,32,36,40], and authors describe linguistic, cognitive, and neuroanatomical correlations. According to these studies, patients with language disorders also perform worse in tests assessing EFs.

### 4.3. svPPA and Control Groups

Our results show only one significant difference between these two groups for RFFT design fluency (*p* = 0.030). The literature indicates that svPPA patients present few executive difficulties at the onset of the disease [29,41]. More specifically, verbal and visuospatial working memory [3,32], inhibition [39], cognitive flexibility [29], initiation [28], and planning [36] are found to be preserved. Although several tests assessing EFs involve language, patients with svPPA do not differ significantly from controls. The RFFT design fluency task differs from the others, as the svPPA group obtained significantly lower results than the control group (*p* = 0.030). Indeed, some patients had difficulties with this task, which however does not involve language or semantic memory, unlike a verbal fluency test. These results indicate that patients may exhibit a mild initiation deficit, linked to frontal lobe dysfunction [17]. Thus, the design fluency task highlights the presence of subtle initiation difficulties at the onset of the disease. To conclude, the svPPA group presented few EF significant difficulties. These results show that cognitive functions that involve little language or conceptual knowledge (semantic memory) are relatively preserved at the onset of svPPA. 

Overall, these results highlight an executive dysfunction in the logopenic and non-fluent variants of PPA but few executive difficulties in the semantic variant [68]. This could be linked to the neuroanatomical regions implicated in PPA [20]. Posterior fronto-insular atrophy exists in the non-fluent variant and temporoparietal atrophy in the logopenic type. EFs are mostly subserved by the frontal lobes, but parietal association areas have connections to the frontal cortex [48]. An overlap exists between the language processing pathways and those subserving EFs [17]. As for the semantic variant, although it belongs to the FTLD group [3,69], temporal involvement seems to predominate, and thus this variant appears to be less vulnerable to frontal atrophy and executive difficulties.

### 4.4. AD and Control Groups

The AD group showed poorer results than the control group for all the tasks assessing EFs. These results are consistent with previous studies describing significant executive dysfunction in AD [13,31,32,43,47,48,70,71]. The profile is characterized by impaired flexibility, inhibition, initiation, and planning [15,70].

### 4.5. lvPPA and nfvPPA Groups

The lvPPA and nfvPPA groups’ results were significantly different for visuospatial short-term and working memory, evaluated by the forward (*p* < 0.0001) and backward (*p* = 0.020) visual spans, results on which were lower for the nfvPPA group. Thus, few of the EFs that were assessed differed between the nfvPPA and lvPPA groups. These results support recent studies by identifying few tasks that significantly differentiate between these two groups [36,72]. Verbal short-term memory (*p* > 0.0001) is similarly impaired in both groups, as described in the literature [24,27,36,72]. However, these data are controversial [16], as one study describes this function as more impaired in nfvPPA patients than in lvPPA patients. Verbal working memory, evaluated by the backward digit span, showed a significant difference (*p* = 0.21) between the two groups, with lower scores for the nfvPPA group, as described in another recent study [72]. Regarding processing speed (TMT A) and cognitive flexibility (TMT B), the results show no significant difference between the two groups, which is consistent with the literature [27,32,36]. The results of the Tower of London test, which assesses planning abilities, showed no significant difference between the two groups, supporting a recent study that observed similar outcomes [36]. These results show that lvPPA patients present the same level of difficulty as nfvPPA patients in performing processing speed and cognitive flexibility tasks, but also planning, a high-level function, as shown in Diamond’s model [16]. The impaired activation of frontal lobes can be considered [19]. Visuospatial short-term memory and working memory are weaker in individuals with nfvPPA than lvPPA, which was not found by another recent study [40]. According to the authors, the poorer performance in lvPPA might be due to bilateral temporoparietal atrophy. In our study, the logopenic patients may have greater frontotemporal atrophy than right parietal involvement, as also described in the literature [36]. A subgroup of lvPPA patients, for whom visuospatial spans showed no significant difference from the control group, presented greater frontotemporal atrophy than parietal involvement. These results suggest an impairment that extends into the nonverbal domain, even at low levels of difficulty [40]. Consistent with Diamond’s model [16], EFs are not altered in equivalent ways in these two variants of PPA. Verbal working memory, inhibition, and initiation are more deficient in nfvPPA than in lvPPA.

### 4.6. lvPPA and svPPA Groups

Processing speed (TMT A) (*p* < 0.0001), verbal short-term memory (*p* = 0.007), verbal working memory (*p* = 0.010), and cognitive flexibility (TMT B) (*p* = 0.005) differentiated the two groups. These results corroborate recent studies that found a significant difference between these two PPA variants, specifically for short-term and auditory–verbal working memory [27,68,72,73]. Visuospatial short-term memory and working memory are preserved in both groups, as described in several studies that found no significant difference [27,42]. Initiation did not show a significant difference either, also as described in a recent study [27]. The results show no difference between the two groups for the tasks assessing inhibition (Stroop interference) and planning (Tower of London). These functions, which appear to be preserved in svPPA, did not appear significantly impaired in lvPPA. Matias-Guiu et al.’s study also found no significant difference between the two groups for these functions [36]. These authors explain this by the fact they only included patients at the onset of the disease. To conclude, few tests assessing EFs distinguished between the lvPPA and svPPA groups.

### 4.7. lvPPA and AD Groups

In comparison to the lvPPA and AD groups, only visuospatial short-term and working memory, evaluated by forward and backward visual spans, respectively, distinguished between the lvPPA and AD groups (*p* < 0.0001). Moreover, it is interesting to note that this study highlighted preserved verbal short-term memory (assessed by the forward digit span) in AD but an impairment in lvPPA, which may contribute to the differential diagnosis of these disorders. Integrating visuospatial span tests into assessments would be relevant since they make it possible to distinguish between patients with lvPPA and those not only with AD but also with nfvPPA [40]. The patients with lvPPA did not demonstrate deficits in visuospatial memory (assessed by visuospatial span tasks). They differed in this way from the AD group, which did show a deficit. These results could be explained by less diffuse atrophy in the temporoparietal junction, which has been found in studies reporting that disorders are weaker at the onset of the disease, due to limited atrophy [56]. It would be interesting to meet patients one year later to find out whether they have developed visuospatial short-term and working memory disorders, suggestive of AD. Teichmann et al., who studied lvPPA with a cohort of 19 patients, concluded that this syndrome was a “logopenic aphasia complex”, showing rapid changes in cognition and brain damage [55].

### 4.8. Other Between-Group Differences

A comparison of the nfvPPA and svPPA groups demonstrated significant differences between these two groups for all the tests assessing EFs, with the nfvPPA group scoring lower (*p* < 0.05). These results are in line with the literature, which states that EFs are relatively preserved in svPPA and impaired in nfvPPA [41]. Recent studies have revealed significantly lower performance in patients with nfvPPA than with svPPA, especially for verbal short-term and working memory [23,24,27,36], processing speed (TMT A), and cognitive flexibility (TMT B) [74]. Contrary to our results, inhibition (Stroop interference) and planning (Tower of London) were not found to be significantly different in the two groups according to some studies [32,36], and nor was initiation (design fluency) [27]. To our knowledge, only a few studies so far have compared short-term and visuospatial working memory between the two groups [31,32,40]. Therefore, our study provides new information for the differential diagnosis of these two forms of PPA.

Regarding the nfvPPA and AD groups, on the forward and backward digit spans, which assess verbal short-term and working memory, respectively, the nfvPPA group scored significantly lower than the AD group. A recent study showed a significant difference between these two groups for verbal short-term memory [72]. These results are not in agreement with studies that revealed no significant difference between the two groups for digit spans [23,74]. Language contamination (i.e., the level of expressive language impairment) in part could help to explain these inconsistent findings.

Finally, analyses of the svPPA and AD groups showed significantly lower performance in the AD group than the svPPA group for all EFs, except verbal short-term memory. Thus, these results provide additional information to help differential diagnosis and support the data from the literature. For instance, verbal working memory, flexibility, inhibition, initiation, and planning are all less efficient in AD than in the semantic form of PPA [23,32,37,40,72,73,75,76]. Only verbal short-term memory does not differentiate the two groups [37,73,76].

Overall, for the in-between group analyses, the nfvPPA and svPPA patients differed the most on EF tests, followed by the svPPA and lvPPA groups. The nfvPPA and lvPPA groups did not differ significantly on the EF tests. 

Although patients essentially express a complaint related to language, executive disorders are observed from the onset of PPA [9,41]. Recent studies highlight the importance of an exhaustive cognitive assessment, not limited to the language domain, but including EFs [8,77]. The consensual diagnostic criteria proposed by international researchers and clinicians [3] constitute a key reference. However, due to frequent classification difficulties, several authors have wondered whether the diagnostic criteria should be revised or made less restrictive [78]. Indeed, some authors propose a revision of the criteria [7]. Others suggest that less restrictive diagnostic criteria would be relevant for each variant of PPA [79]. The results of this study are consistent with current studies, suggesting that the criteria can be retained but should be enhanced with references to executive features to clarify the diagnosis [41,53]. 

Our study has some limitations. The main limitation was the small sample size in each group, which could explain that the findings are at odds with some studies, especially for executive functioning in nfvPPA and lvPPA groups. Other factors including disease severity, language contamination, and the inherent variation at the individual level account for the varying findings across studies. The statistical analysis containing non-parametric tests has restricted the generalization of our results. Another limitation was the lack of qualitative analyses. It would have been interesting to study the types of errors made in the tests and compare them in the different groups. We plan to conduct this qualitative analysis. Another limitation is that only the cognitive domain was studied. An evaluation of the behavioral and social components would have provided additional information. Finally, a longitudinal study would have been valuable. Future research is warranted to provide qualitative analyses.

## 5. Conclusions

This study has provided new insights and led to a better characterization of lvPPA. Patients evaluated at the early stage of the disease showed the moderate impairment of EFs, with the preservation of inhibition and visuospatial short-term and working memory. In comparison with the lvPPA and AD groups, only the forward and backward visual spans distinguished the groups. Visuospatial span task performance is preserved in lvPPA patients, unlike the AD group, which showed a deficit. Moreover, the preservation of forward digit span performance in AD and impairment in lvPPA were also observed, which may contribute to the differential diagnosis of these two disorders. Integrating visuospatial span tests in the cognitive assessment of PPA would be relevant since they make it possible to distinguish lvPPA patients from both AD and nfvPPA patients. 

To conclude, the findings support other recent studies and suggest that, although language deficits remain the core symptoms of PPA, executive dysfunction is also observed at the early stages, even though some studies have described EFs as remaining relatively unaffected, and they are currently excluded from the consensual diagnostic criteria [3,41]. Dysfunctions specific to each variant of PPA were identified. For instance, patients with nfvPPA presented the most dysexecutive profile, followed by those with lvPPA, and then svPPA. Tasks such as the Stroop test, visuospatial spans, and RFFT design fluency could enhance assessments. Indeed, as recently described in the literature, nonverbal tests, specifically design fluency and visuospatial spans, help to distinguish the different variants of PPA and AD [40]. Our results confirm Macoir et al.’s [41] point of view, which suggests that knowledge of PPA should be updated based on this dysexecutive semiology. Thus, we propose to consider high-level cognitive functions such as EFs in the assessment of PPA to assist with accurate diagnoses and plan the best non-pharmacological treatment as soon as possible. A rigorous and thorough assessment, both linguistic and executive, may contribute to the differential diagnosis of lvPPA, as opposed to the two other types and also AD. 

Finally, our research highlights a need for standardized tools that can be used for studying the complex aspects of EFs, so that disorders that may go unnoticed at the onset of the disease can be accurately diagnosed [80,81].

## Figures and Tables

**Figure 1 brainsci-13-00406-f001:**
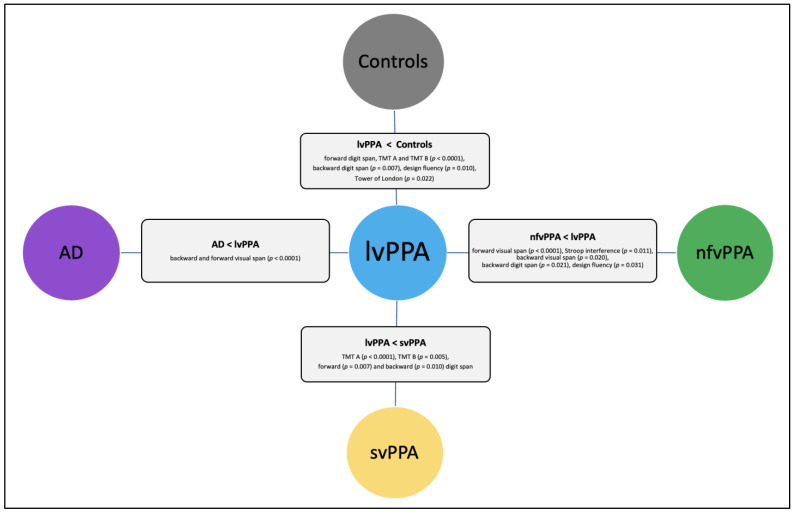
Executive tasks showing statistically significant differences between lvPPA and other clinical groups, inspired by Matias-Guiu et al. [36].

**Table 1 brainsci-13-00406-t001:** Demographic information for controls and PPA groups.

	Controls (*n* = 41)	lvPPA (*n* = 22)	nfvPPA (*n* = 22)	svPPA (*n* = 26)	AD (*n* = 32)	F	*p*-Value
Gender (M/F)	13/28	13/9	7/15	8/18	18/14	7.612 ^1^	0.124
Handedness (R/L)	39/2	20/2	21/1	26/0	32/0		
Age: mean (SD) Minimum/maximum	65.73 (±10.1) 41/86	68.95 (±9.3) 46/84	69.55 (±4.8) 57/78	66.27 (±10.2) 48/83	68.66 (±9.5) 53/85	1.458	0.285
Education: mean (SD)	12.07 (±3.7)	11.59 (±2.5)	11.68 (±2.8)	12.42 (±3.5)	11.69 (±2.7)	1.667	0.183
Disease duration months (SD)	/	26.45 (±3.54)	27.00 (±4.44)	27.46 (±4.85)	25.31 (±5.85)	1.542	0.194
MMSE/30 (SD)	29.22 (±0.9)	26.05 (±1.8) ^a,c^	22.64 (±5.1) ^a,b^	26.58 (±3.6) ^b,d^	22.75 (±2.9) ^c,d^	82.930	< 0.001

Note. M = male; F = female; R = right; L = left; SD = standard deviation; MMSE = Mini-Mental State Examination; ^1^ = chi-squared test; significant group differences on MMSE performance between groups (*p* < 0.05): ^a^ *nfvPPA* < lvPPA; ^b^ *nfvPPA* < svPPA; ^c^ *AD* < lvPPA; ^d^ *AD* < svPPA.

**Table 2 brainsci-13-00406-t002:** Mean of raw scores (and standard deviation) and comparison of groups in tests assessing executive functions.

	Controls *n* = 41	lvPPA *n* = 22	nfvPPA *n* = 22	svPPA *n* = 26	AD *n* = 32	Mann-Whitney Tests
**Forward digit span (SD)**	6.05 (1.32)	4.57 (0.97)	4.56 (1.01)	5.50 (1.01)	5.20 (1.78)	lvPPA, nfvPPA < C **; AD < C *; nfvPPA < AD *; nfvPPA < svPPA **; nfvPPA < lvPPA *; lvPPA < svPPA *
**Backward digit span** **(SD)**	4.39 (1.35)	3.71 (1.25)	3.00 (0.70)	4.71 (1.63)	4.20 (1.09)	nfvPPA < C **; AD < C *; nfvPPA < AD *; AD < svPPA *; nfvPPA < svPPA **; nfvPPA < lvPPA *; lvPPA < svPPA *
**Forward visuospatial span (SD)**	5.90 (0.83)	5.43 (0.53)	4,22 (0.66)	5.79 (0.89)	3.80 (0.44)	nfvPPA, AD < C **; AD < svPPA **; AD < lvPPA **; nfvPPA < svPPA **; nfvPPA < lvPPA **
**Backward visuospatial span (SD)**	4.85 (1.15)	4.43 (0.87)	3.89 (1.05)	5.14 (0.77)	3.20 (1.64)	AD < C **; nfvPPA < C *; AD < svPPA **; AD < lvPPA **; nfvPPA < svPPA **; nfvPPA < lvPPA **; nfvPPA < lvPPA *
**Stroop interference** ** (s)(SD)**	128.90 (25.96)	197.00 (135.13)	322.33 (149.63)	130.79 (33.18)	218.80 (42.92)	nfvPPA, AD < C **; AD < svPPA *; nfvPPA < svPPA **; nfvPPA < lvPPA *
**TMT A** **(s) (SD)**	41.12 (14.05)	68.71 (24.24)	83.89 (39.73)	43.29 (10.02)	78.00 (20.62)	lvPPA, nfvPPA, AD < C **; AD < svPPA **; nfvPPA < svPPA **; nfvPPA < lvPPA *; lvPPA < svPPA **
**TMT B (s)** **(SD)**	96.24 (31.47)	190.86 (103.15)	274.67 (113.09)	119.86 (75.13)	165.80 (49.73)	lvPPA, nfvPPA, AD < C **; AD < svPPA *; nfvPPA < svPPA **; nfvPPA < lvPPA *; lvPPA < svPPA *
**RFFT** **No. of Prod. (SD)**	73.39 (20.81)	60.43 (24.86)	33.33 (18.77)	64.14 (20.67)	41.40 (37.95)	nfvPPA, AD < C **; lvPPA, svPPA < C *; AD < svPPA*; nfvPPA < svPPA **; nfvPPA < lvPPA **; nfvPPA < lvPPA *
**Tower of London** **No. Mvt. 3 N (SD)**	3.06 (0.19)	3.18 (0.25)	3.33 (0.33	3.04 (0.11)	3.40 (0.59)	lvPPA, nfvPPA, AD < C *; nfvPPA < svPPA *
**Tower of London** **No. Mvt. 5 N (SD)**	5.91 (1.32)	8.33 (5.91)	11.77 (2.90)	6.71 (2.70)	8.40 (4.12)	nfvPPA < C **; nfvPPA < svPPA *; nfvPPA < lvPPA **; nfvPPA < lvPPA *
**Tower of London** **No. Mvt. 5 i+ (SD)**	5.46 (1.31)	5.80 (1.16)	7.88 (5.83)	6.11 (2.41)	8.66 (4.57)	AD < C **; lvPPA < C *; AD < svPPA *
**Tower of London** **No. Mvt. 5 i- (SD)**	7.02 (2.25)	11.71 (5.68)	14.92 (7.71)	7.35 (2.62)	14.33 (5.12)	lvPPA, nfvPPA, AD < C *; AD < svPPA *; nfvPPA < svPPA *

Note. SD = standard deviation; s = second; No. Prod. = number of productions; No. Mvt. = number of movements; C = controls; ^*^
*p* < 0.05; ^**^ *p* < 0.0001.

## Data Availability

Data supporting reported results are available from the corresponding author on request.

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
