# Peer review of "Executive Profile of the Logopenic Variant of Primary Progressive Aphasia: Comparison with the Semantic and Non-Fluent Variants and Alzheimer’s Disease"

_brainsci, 2023, doi:10.3390/brainsci13030406_

Round 1

Reviewer 1 Report

This study investigated the executive functioning (EF) profiles of the primary progressive aphasia (PPA) variants and Alzheimer’s disease (AD), with a particular emphasis on evaluating the EF profile of the logopenic variant of PPA (lvPPA), relative to the other groups. The study is interesting, the sample sizes are adequate, and the manuscript is relatively well-written. Nevertheless, I have some comments and concerns, listed below:

Major concerns

i) I appreciate that the authors have attempted to translate the study findings to the clinical setting by creating an ‘algorithm inspired by Marshall et al.’s road map’ (Figure 2). I have a few concerns about this figure and its interpretation and wonder if its inclusion in the paper adds value. Specifically:

-       If the PPA syndrome can be distinguished based on the language profiles alone (as is demonstrated by Marshall et al.’s language map), then what relevance is there in testing EF? 

-       I find the EF component of the road map too simplistic. For example, the nfvPPA arm of the road map states, ‘is there a dysexecutive syndrome?’. If the answer was ‘no’, would this mean the patient does not have nfv-PPA? Further, does this imply that the nfvPPA group is the only group with a dysexecutive syndrome? This is at odds with the reported study findings (the abstract states: ‘Thus, this research highlighted the existence of an executive dysfunction from the onset of the disease in lvPPA but also in the other two variants of PPA’).  

-       Similar concerns relate to the other arms of the road map. For example, to the major decision ‘are visual-spatial memory and inhibition preserved?’, if the response was ‘no’, then what syndrome would the patient have?

-       I question if the word ‘algorithm’ is misleading. In my opinion, an algorithm typically refers to instructions derived from mathematics or computer science. If the authors deem an EF algorithm is important, then they could consider creating an EF algorithm, such as a decision tree or random forest classifier, using the existing dataset. These analyses can be easily created in many free, open access software (e.g., JASP, Jamovi etc.). If the authors deem algorithm analyses go beyond the scope of this study, then the authors could consider performing ROCs in the Supplementary Material (i.e., to statistically determine the most sensitive executive measures at distinguishing PPA groups) to support their road map claims.

-       Disagreement still exists in the literature regarding the PPA EF profiles. For example, Coeman et al.’s (2022; https://doi.org/10.1016/j.cortex.2022.10.001) meta-analysis found that EF was similarly affected in lvPPA and nfvPPA. In my opinion, the road map’s claims are too audacious based on a study of only 70 PPA patients.

The authors may consider revising the road map and/or downplaying its use by putting it in the Appendix or Supplementary Material of the manuscript (rather than in the main text).

ii) Figure 1. The figure does not demonstrate the group differences between lvPPA and svPPA. For example, lvPPA < svPPA on TMT A and TMT B. I appreciate that the authors have tried to communicate the group comparison findings in a visually appealing way; however, I question if this is the best way to present this information. Notably, the study’s main aim is to evaluate the lvPPA EF profile. Perhaps the lvPPA group could be centred in the middle, and the figure only displays comparisons with lvPPA? If done this way, then a small advantage is the control group could be included. The authors may consider other ways to improve communication of the group findings.

iii) I have some overall concerns which could probably be addressed in the limitations section of the paper. The study found that the nfvPPA group demonstrated widespread executive dysfunction relative to the other groups. These findings, however, are at odds with Coeman et al.’s (2022) meta-analysis, which found that executive functioning was similarly affected in lvPPA and nfvPPA. Could factors including disease severity and/or language contamination (for measures that require language proficiency, e.g., Digit Span, Stroop) account for the varying findings across studies? This study does not include information on the language and/or functional capacity profiles of the PPA variants, and so these queries cannot be directly addressed. For me, this is a limitation.

Relatedly, is information regarding the prevalence of parkinsonism in the nfvPPA group available? Patients with nfvPPA typically develop motor apraxia, and a proportion of these patients go on to develop a frank Parkinson’s Plus syndrome (Ulugut et al. 2022 https://doi.org/10.1007/s00415-021-10689-1). These impairments can contribute to distinct cognitive and functional problems. If parkinsonian data is unavailable, then this should be acknowledged in the limitations paragraph of the manuscript.

Minor concerns

Table 2. Why has the author put the AD group in between the lvPPA and nfvPPA groups? Would it be more suitable to put the PPA variants together? Further, is there a reason why group comparisons with the control group were not included in the table? Lastly, the authors may consider displaying the table across the breadth of the page (rather than indexed to the right side of the page). It may help with the line spacing. 

Lines 200-225 are under the heading ‘3.2. Group comparison to controls’. This paragraph should have its own heading (e.g., ‘3.3. Patient group comparisons’ or ‘Comparing the clinical groups’ etc.).

Lines 375-377. The authors claim, ‘To our knowledge, no study so far has compared short-term and visuospatial working memory between the two groups (nfvPPA and svPPA). Therefore, our study provides new information for the differential diagnosis of these two forms of PPA.’ Foxe and colleagues, however, specifically investigated these skills in two studies (Spatial Span: https://doi.org/10.1016/j.cortex.2020.08.018; Box Task and Spatial Span: https://doi.org/10.1111/ene.15035). These papers should be acknowledged.

Lines 378-382. The authors found that the nfvPPA group performed worse than the AD group on the digit span tasks. The authors, however, report other studies which found no such difference. Could language contamination (i.e., the level of expressive language impairment) in part contribute to these inconsistent findings?

Author Response

Dear Reviewer,

Best regards,

Sandrine Basaglia-Pappas

Reviewer 2 Report

This is a very interesting study, presenting significant results for the investigation of neuropsychological deficits in different PPA variants. I appreciate and acknowledge the importance of presenting such evidence, as previous studies are scarce. The rationale of the study is justified however there are some issues, I address in detail below:

Methods

Due to the small sample size in each group, it would be suggested to use Shapiro-Wilk instead of Kolmogorov-Smirnov test to explore normality (although I would not expect results to be extremely different).

Results

Please include p values for demographics comparisons in Table 1. Moreover, please specify which test did you use for gender (Kruskal-Wallis and Mann-Whitney tests are not applicable).

Table 2 is relatively confusing. You could formulate it presenting the results for each comparison in a more comprehensive way. Moreover, it would be very helpful if the authors included the error bar charts for all comparisons. 

Discussion

Line 247-248: The authors attributed not pathological performance of the lvPPA group 

in inhibition tasks on the fact that “patients were recruited at the onset of the disease”. However, this is in contrast with demographics for the lvPPA group 26.45 (±3.54). Perhaps, the authors could further discuss their findings within the framework of contemporary studies. 

In general, results discussion could be further expanded for each set of comparisons. 

Moreover, although I appreciate the difficulty to conduct this study and the great importance of presenting such results, I would suggest the authors to be more modest with regard to the algorithm they suggest and the novelty of their results (for instance in conclusion). The authors should acknowledge the fact that each cohort included a relatively small sample of patients and the statistical analysis contained non-parametric tests that restricts generalization of their results.

Author Response

(The authors gave the same response as above.)

Round 2

Reviewer 1 Report

The manuscript has improved considerably.

Minor comments

The diagnostic algorithm / roadmap is no longer referred to in the main text. The authors should revise lines 26-27 of the abstract.

Lines 54-56. Consider revising to ‘In this clinical syndrome, neuropathology is heterogeneous, with 60% of patients with PPA presenting with frontotemporal lobar degeneration (FTLD), and the remaining 40% with Alzheimer’s disease (AD) pathology’.

Lines 60-61. There is some confusion whether the authors are referring to the clinical syndrome Alzheimer’s disease or patients with underlying Alzheimer pathology (for example, some patients with corticobasal syndrome have underlying Alzheimer pathology but do not reach diagnostic criteria for ‘AD’). Consider clarifying to improve the main message of lines 60-65.

Table 1. Were the controls not statistically different from the patient groups on the MMSE? Further, were there statistically significant differences between the patient groups on the MMSE? MMSE F and p value should be included in Table 1. Significant group differences could be reported in the notes section below the Table.

The authors have revised Figure 1. The authors should now remove the statement “well as other between-group comparisons.” [Lines 244-245] 

Author Response

Dear Reviewer,

Best regards, 

Sandrine
